# A Deep Learning Model for Fault Diagnosis with a Deep Neural Network and Feature Fusion on Multi-Channel Sensory Signals

**DOI:** 10.3390/s20154300

**Published:** 2020-08-01

**Authors:** Qing Ye, Shaohu Liu, Changhua Liu

**Affiliations:** 1School of Computer Science, Yangtze University, Jingzhou 430023, China; 2School of Mechanical Engineering, Yangtze University, Jingzhou 430023, China; liushaoh@126.com; 3General Office, Yangtze University, Jingzhou 430023, China; yangtzelch@163.com

**Keywords:** array signal processing, feature fusion, deep neural network, multi-channel sensory signals, intelligent fault diagnosis

## Abstract

Collecting multi-channel sensory signals is a feasible way to enhance performance in the diagnosis of mechanical equipment. In this article, a deep learning method combined with feature fusion on multi-channel sensory signals is proposed. First, a deep neural network (DNN) made up of auto-encoders is adopted to adaptively learn representative features from sensory signal and approximate non-linear relation between symptoms and fault modes. Then, Locality Preserving Projection (LPP) is utilized in the fusion of features extracted from multi-channel sensory signals. Finally, a novel diagnostic model based on multiple DNNs (MDNNs) and softmax is constructed with the input of fused deep features. The proposed method is verified in intelligent failure recognition for automobile final drive to evaluate its performance. A set of contrastive analyses of several intelligent models based on the Back-Propagation Neural Network (BPNN), Support Vector Machine (SVM) and the proposed deep architecture with single sensory signal and multi-channel sensory signals is implemented. The proposed deep architecture of feature extraction and feature fusion on multi-channel sensory signals can effectively recognize the fault patterns of final drive with the best diagnostic accuracy of 95.84%. The results confirm that the proposed method is more robust and effective than other comparative methods in the contrastive experiments.

## 1. Introduction

Final drive is the core component in the rear axle of automobile and is always running under complex operating conditions and inevitably faults. Any fault of final drive may cause severe human injury, production stoppage and economic loss [1,2]. The change of status is hidden in the vibrational characteristics produced during the operating period. Therefore, fault diagnosis with vibration signal is extremely efficient in status detection and failure recognition of machinery [3,4,5,6,7,8]. P.K. Wong used Ensemble Empirical Mode Decomposition (EEMD) in the failure recognition of gears [9]. Stephen McLaughlin acquired gear features from modulating signal [10]. D.P. Jena et al. used variant technology of wavelet transform to implement fault diagnosis [11]. M Amarnatha utilized vibration signals to detect the severity of fault of a helical gear tooth [12].

Diagnosing fault using a single-channel vibration signal may cause the decrease of diagnostic accuracy due to the direction of sensor and installation position [13]. Array signal processing is extensively applied in sonar, wireless communications, medical diagnosis and engineering applications [14,15,16,17]. Array signal processing could enhance useful signal and suppress noise and interference by placing a set of sensors in different positions to form sensor array [18,19,20,21]. With the purpose of improving the diagnostic accuracy, it is essential to obtain multi-channel sensory signals by setting some sensors along different directions. The diagnostic accuracy could be more superior and reliable than using single signals.

Nowadays, many researches contributed to information mining and the utility of massive data [22]. Moreover, vibrational signals collected from multiple sensors are widely used in the field of fault diagnosis [23,24,25,26]. Yaguo Lei used multi-sensor data in fault detection of gearbox [27]. M.S. Safizadeh studied multi-sensor data fusion to improve the performance of fault recognition for rolling element bearings [28]. Luyang Jing combined deep neural networks and multi-sensor data fusion in the fault detection of planetary gearbox [29]. Zhixiong Li researched in fault recognition of gear cracks with multi-channel sensory signals [30]. Joao A. Duro adopted multi-sensor signals in machinery condition monitoring [31]. Khazaee M. adopted a feature-level fusion in fault diagnosis of planetary gearbox [32]. Yunusa-Kaltungo developed a novel framework of fault characterization for rotating machines [33]. Zhiwen Liu used support vector machine and ant colony into fault diagnosis of gearbox [34]. Previous researches usually adopted traditional machine learning technologies in machinery fault diagnosis. Few papers have studied the deep feature learning and fusion model with multi-channel data.

The vibration signals are unavoidably polluted by disturbance originated from other vibration components and paroxysmal noise [35,36]. Traditional fault diagnosis employs complex signal process techniques to extract the shallow features manually. However, the representativeness of features obviously affects the capability of traditional fault recognition. Moreover, the manual selection of most sensitive features for a specific issue is time consuming [37]. These shortcomings encourage researchers to find out a new method to adaptively and automatically learn deep features from original data. 

After a series of non-linear transformations, representative and essential features can be effectively mined by deep learning. Deep learning includes three basic types: Convolutional Neural Network (CNN), Deep Neural Network (DNN) and Deep Belief Network (DBN) [38,39,40,41]. Recently, deep learning using deep architectures is widely applied for intelligent fault diagnosis [42,43,44]. Compared with DBN and CNN, DNN consisted of a stack of auto-encoders that is purely unsupervised and easy to implement [45,46]. Due to the deep architectures, DNN can mine fault-sensitive features from raw signals so as to effectively ferret out the non-linear relationship between symptoms and faults. Jia Feng applied deep learning technology into the field of rotating machinery [37]. J. Zabalza et al. employed stacked auto-encoder to extract feature in hyper spectral imaging [47]. P. Xiong et al. employed de-nosing auto-encoder for enhancement of signal [48].

Despite the fact that deep features extracted from multi-channel sensory data are fault sensitive and representative, features of different channels are usually heterogeneous and redundant. In order to avoid performance reduction caused by information redundancy, Locality Preserving Projection (LPP) is used in the fusion of extracted features obtained from multi-channel sensory signals. LPP is a typical dimensionality reduction algorithm based on local structure of data which can effectively preserve the neighborhood information and local characteristics during the fusion procedure [49,50,51,52]. The fused deep features of multi-channel sensory data will be fed into a diagnostic model based on softmax [53,54,55].

In this paper, aiming at solving the limitation of single sensor and automatically extracting fault-sensitive features without manual pre-processing, a deep learning model with multi-channel sensory signals is proposed and verified in application of intelligent fault diagnosis of automobile final drive. The contrast analysis and experimental results show that the proposed method can adaptively extract deep features from original multi-channel signals and effectively improve the accuracy and capability of failure recognition. The contributions and key techniques of our work are summarized in three aspects:(1)To solve the limitation of single sensor, we collect multi-channel sensory signals by installing several sensors in different positions along horizontal and vertical direction so as to implement reliable monitoring.(2)To solve the limitation of using traditional signal process techniques to manually extract features, we employ deep learning technique in learning representative features from original multi-channel signal adaptively.(3)In order to avoid the heterogeneity and redundancy of deep features from multi-channel data, we fuse these deep features by using locality preserving projection.

The following is organized in four sections. In Section 2, the fundamental theory used in the research is given. Application of the diagnostic method based on multiple DNNs (MDNNs) is presented in the next section. A set of experiments is implemented to evaluate the accuracy and efficiency in Section 4. In Section 5, a set of conclusions is provided.

## 2. The Fundamental Theory of Auto-Encoder

An auto-encoder is a special neural network based on unsupervised learning with three necessary layers of artificial neural network. The training process of an auto-encoder always contains two stages, encoding and decoding [56].

Given a sample set X = {x^1^,…, x^i^,…, x^M^}, x^i^
∈ R^N^, where M denotes the sample size. In encoding stage, input data $x^{i}$ is transformed into a low-dimensional feature space to learn an approximation of input data as shown in below:(1)$$h\left(i\right)=f_{\theta }\left(x^{i}\right)$$
where $f_{\theta }$ denotes encoding function and θ=[W,b] denotes the parameters between the first layer and the second layer.

In decoding stage, input data $x^{i}$ can be reconstructed in output layer as follows:(2)$$x^{i\prime }=g_{\theta^{\prime }}\left(h\left(i\right)\right)$$
in which $g_{\theta^{\prime }}$ denotes decoding function, $x^{i\prime }$ denotes the reconstruction of original data $x^{i}$, and $\theta^{\prime }=\left[W^{\prime },b^{\prime }\right]$ is the parameters between the second layer and the third layer.

By minimizing the average reconstruction error of M samples, the parameter set $\left\{\theta ,\theta^{\prime }\right\}$ of the encoding stage and decoding stage can be optimized:(3)$$\emptyset_{\mathrm{AE}}\left(\theta ,\theta^{\prime }\right)=\frac{1}{M}\sum_{i=1}^{M}L\left(x^{i},x^{i\prime }\right)$$
(4)$$L\left(\;x,x^{\prime }\right)=\|x-{x^{\prime }\|}^{2}$$
where $L\left(\;x,x^{\prime }\right)$ represents reconstruction error function to measure the discrepancy between original data and its reconstruction.

## 3. Proposed Diagnostic Model

A deep learning method with deep architecture and feature fusion on multi-channel sensory signals is developed in this paper. It includes three parts: the construction of deep architecture for feature learning, the fusion of the deep features extracted from multi-channel sensory data and the construction of intelligent diagnostic model using softmax.

### 3.1. Construction of Deep Neural Network for Deep Feature Learning

By piling up N auto-encoders, construct a DNN with N hidden layers so as to hierarchically learn essential characteristics from sensory data. The first auto-encoder consists of input layer and the first hidden layer, and so on. The training of DNN contains two sections: pre-training and fine-tuning [57,58,59]. During the process of the first section based on unsupervised learning, the encoder vector of x(i) obtained from the first auto-encoder is:(5)$$h{\left(i\right)}^{1}=f_{\theta^{1}}\left(x\left(i\right)\right)$$
where $\theta^{1}$ represents the parameter of the first auto-encoder.

Then, the first encoder vector $h{\left(i\right)}^{1}$ is inputted into the second auto-encoder, which is consists of the first hidden layer and the second hidden layer, and so on. In this way, the Nth encoder vector of x(i) is obtained as follows:(6)$$h{\left(i\right)}^{N}=f_{\theta^{N}}\left(h{\left(i\right)}^{N-1}\right)$$
where $\theta^{N}$ is the parameter of the Nth auto-encoder.

Compared with random parameters, by minimizing the reconstruction error, the unsupervised pre-training of DNN can improve the generalization of parameters. In the process of fine-tuning parameters backwards based on supervised learning through the BP (Back Propagation) algorithm, the ability of feature learning is further enhanced. BP algorithm compares the output of output layer with the corresponding label so as to calculate the loss value. The loss function is shown below:(7)$$\emptyset_{\mathrm{DNN}}\left(\theta \right)=\frac{1}{M}\sum_{i=1}^{M}L\left(x\left(i\right),\;h{\left(i\right)}^{N}\right)$$
where $\theta =\left\{\theta^{1},\theta^{2},\ldots ,\theta^{N}\right\}$ corresponds to N hidden layers. The partial derivative of the parameter is solved, and then the parameter is updated by the gradient descent algorithm. The parameter set can be optimized with learning rate of μ as follows:(8)$$\theta =\theta -\mu \frac{\emptyset_{\mathrm{DNN}}\left(\theta \right)}{\partial \theta }$$

During the training of DNN, a series of non-linear transformations of pre-training captures the local variation of input data, and fine-tuning mines the discriminated information from input data.

### 3.2. Fusion of Deep Features Extracted from Multi-Channel Signal

As shown in Figure 1, with the purpose of obtaining more complete and reliable data, *S* sensors are installed so as to collect multi-channel signals and construct multiple DNNs (MDNNs). In MDNNs, raw timing signals collected from different sensors are inputted into different DNNs. The deep features extracted from MDNNs are usually redundant and high dimensioned. LPP is a dimension-deduction method that can be used to fuse deep features and effectively preserve local structure in lower dimension [60,61].

Let F = {f_1_,…, f_i_,…, f_n_} to represent the deep feature learned from MDNNs, in which f_i_ = $\left[f_{i}^{1},\ldots ,f_{i}^{s},\ldots ,f_{i}^{S}\right]\in R^{D}$, n represents the sample size and S represents the number of sensors. Using a projection matrix $A\in R^{D\times d}$, deep features with a dimension of D are transformed into dimension of d as below:(9)$$f_{i}^{*}=A^{T}f_{i},i=1,\ldots ,n$$

The objective function of LPP is the following formula:(10)$$\min\sum_{\mathrm{ij}}{\left(f_{i}^{*}-f_{j}^{*}\right)}^{2}W_{\mathrm{ij}}$$
(11)$$W_{\mathrm{ij}}=\{\begin{array}{c} \exp\left(-\|f_{i}-f_{j}{\|}^{2}/t\right)\\ 0 \end{array}$$
where $W_{\mathrm{ij}}$ represents the matrix to measure relations of different components in deep feature. With algebraic formulas, transform the objective function into the following formula:(12)$${\min\;A}^{T}{\mathrm{FLF}}^{T}A$$
where L=∅−W is the Laplacian matrix in which ∅ is the diagonal matrix of W.

Then, the above problem with constraint is converted into a generalized eigenvalue problem. The solution of above objective function can be achieved from eigenvectors corresponding to minimum eigenvalues.

### 3.3. The Procedure of Intelligent Diagnostic Model

The fused deep features $F^{*}=A^{T}F$, which can reflect the variation of conditions will be inputted into the following fault classifier based on softmax. As shown in Figure 2, the proposed method with multi-channel sensory data contains two parts: a training diagnostic model based on fused deep features of multi-channel data and fault recognition. The detailed procedure of the proposed method is organized in the following six steps:Step 1:Collect multi-channel sensory data from multiple sensors installed in different directions and positions.Step 2:Without manually extracting features by using traditional signal process techniques, raw data is split up into training subset and testing subset.Step 3:Construct deep architecture with multiple DNNs to learn fault-sensitive and representative features adaptively from multi-channel sensory signals.Step 4:Fuse the deep features learned from MDNNs constructed in Step 3 by using LPP, and acquire the representative low-dimensional features.Step 5:Feed the fused features obtained in Step 4 into the fault classifier based on softmax.Step 6:Implement fault recognition on the testing set to verify the classification ability and generalization of the method.

## 4. Experiments and Discussion

### 4.1. Experimental Arrangement

As shown in Figure 3, a set of contrastive experiments are implemented on the test rig of automobile final drive. The test rig contains a control cabinet section, a drive section to activate the driving motor and a fixture section. The rotating speed can be controlled by the cabinet section. The fixture section is to simulate the running situation under a specific rotating speed. To carry out reliable monitoring, two vibration acceleration sensors are located in orthogonal positions so as to collect more stable multi-channel sensory data along a different transmission path. This kind of sensor is frequently used to acquire high-frequency signals in mechanical vibration engineering. The characteristics of vibration acceleration sensors are a wide frequency response, a wide dynamic range, and high reliability. Figure 4 shows the locations of multiple sensors.

The most common failure modes of final drive include gear crack, gear error, gear tooth broken, gear burr, misalignment and gear hard point [12]. These faults usually occur in the gears pair as shown in Figure 5 and Figure 6. Signals from the above 6 fault modes and normal mode are collected. To collect more stable and fault-sensitive vibrational signals, based on previous research and experiences, the frequency of sampling is 12,000 Hz and the rotational speed of the motor is 1200 r/min. The frequency of sampling must be higher than the pre-estimated gear meshing frequency or the detailed information of failure will be lost.

In order to collect enough and reliable data to represent each fault modes, 10 simulations of data acquisition are implemented for each mode. In the processing of data acquisition, the time of duration is 2 s and 25 signals are collected. Each simulation takes 10 s. In this way, 1750 vibration signals corresponding to each different mode are obtained from each sensor. The collected signals are split into training subset with 1400 signals and testing subset with 350 signals. Each vibration signal contains 1024 data points. Descriptions about the final drive fault modes are listed in Table 1.

A series of experiments is performed under Matlab 7.0 on the PC with the configuration of 3.4GHz CPU and 4GB RAM.

To evaluate the superiority of the proposed model, we also utilize BPNN (Back-Propagation Neural Networks) and shallow architecture SVM (Support Vector Machine) to construct diagnostic models for automobile final drive.

To verify the effectiveness of fusing sensory data from multiple sensors, the first set of experiments are implemented between diagnostic models based on multi-channel data and single sensory data. To verify the superiority of deep architecture in feature learning, the second set of experiments is implemented between the proposed method and diagnostic model based on BP neural networks with a shallower structure. To analyze the representativeness of the features learned from hidden layers in deep architecture, the third set of experiments is implemented to reveal most typical features using LPP and Principle Component Analysis (PCA). Ten trials are carried out for each experiment.

### 4.2. Models Design

With regard to a diagnostic model based on MDNNs, the layout of deep architecture has an important impact on accuracy. In order to design the optimal model, we investigate the accuracy variation of different deep architectures. As shown in Figure 7, the diagnostic accuracy improves obviously as the deep architectures changes from containing only one hidden layer to four hidden layers. However, as the hidden layer changes from including 100 neurons to including 1000 neurons with an interval of 100, the diagnostic accuracy is fluctuating up and down. According to the results, the optimal model based on MDNNs consists of three hidden layers and there are 400 units in each hidden layer.

In order to train each DNN, the epoch times in pre-training are set to 100, and the epoch times of fine-tuning are setting to 200 by experience. In the fusion procedure based on LPP, the dimension of fused deep features *d* and hyper-parameter *k* of nearest neighbors are set to 20 and 12 using a cross-validation algorithm.

For the diagnostic model based on BPNN, the shallow architecture contains two hidden layers with 100 units per hidden layer. For the diagnostic model based on SVM, the optimum parameters are chosen by using 10-fold cross validation. The hyper-parameter *C* is set as 10 ^*α*^, in which *α* is in the range of 0 to 2. To achieve non-linear classification, kernel function RBF (Radial Basis Function) with *C* = 10 and *r* = 2 is adopted.

### 4.3. Contrast Analysis and Discussion

#### 4.3.1. Validity of the Fusion Strategy

To verify the validity of the fusion strategy on multi-channel sensory data, we implement the first set of experiments to compare several diagnostic models based on multi-channel data and single sensory data, respectively. Figure 8, Figure 9 and Figure 10 show the detailed diagnostic results of these contrastive models based on the proposed method, BPNN and SVM in ten trials.

The diagnostic results of the proposed method with multi-channel data and single-channel data are 95.8% and 91.4%. It means that the accuracy is improved after the multi-channel data fusion process. The diagnostic results of the model based on BPNN are 79.1%, and the diagnostic results of model based on SVM are 76.6% without fusion of multi-channel data. On closer inspection, the accuracy is increased to 84.6% and 81.3% after fusing the sensory data of multiple sensors.

The comparison result indicates that the multi-channel data collected from multiple sensors are more complete and abundant than single-channel data. Moreover, the fusion strategy can reflect the interdependency between multi-channel data and the fault modes effectively so as to improve the performance of diagnostic model by nearly 8%. In particular, by constructing multiple DNN to learn representative features from multi-channel sensory data and fusing the learned features using LPP algorithm effectively, the diagnostic model using the proposed method exhibits its superiority in fault recognition.

#### 4.3.2. Validity of the Deep Architecture

To verify the superiority of deep architecture in feature learning, this research implements a second set of experiments to compare diagnostic models based on deep architecture and shallow architecture using training set and testing set which are consisted of multi-channel sensory data. Figure 11, Figure 12 and Figure 13 show the differences in classification results of training subset and testing subset among three contrastive diagnostic models based on the proposed method, BPNN and SVM. Each experiment is carried out for ten trials.

As shown in Figure 11, the average results of training subset and testing subset in ten trials for diagnostic model based on MDNNs are 95.8% and 92.8%. As shown in Figure 12 and Figure 13, the average results of training subset for another two models based on BPNN and SVM are 84.6% and 81.3%, and the testing accuracies are 81.1% and 77.6%, respectively.

From Figure 11, Figure 12 and Figure 13, the deep architecture of MDNNs can effectively automatically learn intrinsic features which are sensitive to fault patterns and the testing accuracy of ten trials are very stable in the range of 92.03% to 93.8%. However, the testing accuracy of diagnostic model based on BPNN is unstable and fluctuating from 76.2% to 82.6%. The comparison results indicate that deep feature learning architecture outperforms shallow architecture in feature learning performance on average without using any signal processing and expert experience [24].

#### 4.3.3. Representative of Deep Features

With the purpose of analyzing the representativeness of features learned from the proposed deep architecture with three hidden layers, we implement the third set of experiments to reveal the first two typical features learned from different hidden layers using LPP. Figure 14, Figure 15 and Figure 16 visualize the typical 2-dimensional deep features learned from three hidden layers in the deep architecture.

From Figure 14, Figure 15 and Figure 16, we can find that the first two dimensional features fused from the deeper hidden layer by using LPP are more discrete between different patterns than that of the shallow hidden layer. This demonstrates that the deep architecture can learn more differentiable features from the input data through multiple non-linear transformation layer-by-layer [47]. In particular, the first two dimensional features of the last hidden layer can almost separate samples of each pattern precisely.

For comparison, we preserve the most typical two-dimensional features that are fused by using LPP and Principle Component Analysis (PCA). With the purpose of dimension reduction, we employ PCA in the fusion of deep features learned from the last hidden layer. With the features learned from the last hidden layer with the dimensionality of 400 for each sensor, the output of MDNNs is in the dimension of 2 × 400. By using PCA, the dimensionality of features extracted from MDNNs with multi-channel data is reduced from 2×400 into 20, which is the same dimension of fused features using LPP.

In order to intuitively judge the fault-sensitivity of the learned features, the first two principal components of the fused features are presented for visualization. The typical components fused by PCA of the last hidden layer are shown in Figure 17. Some features of pattern C3 and pattern C5 are mixed and failed to distinguish. By comparing Figure 16 with Figure 17, we can conclude that LPP is more superior in retaining the local structure of input data that is valuable for fault pattern recognition than PCA. LPP achieve fusion and extraction using manifold learning [42,53].

#### 4.3.4. Contrastive Analysis and Discussion

In this study, we mainly focus on the intelligent fault diagnosis with raw vibrational data without previous traditional signal pre-processing. The representative features are automatically extracted using deep structure of networks. However, in the traditional fault diagnosis of machinery, features in time domain and frequency domain can be extracted manually from vibrational signals and be inputted into the fault recognition model [38]. To prove the sensitivity to above two forms of features between different diagnostic models, we compare the proposed model based on MDNNs and diagnostic models based on BPNN and SVM, separately with input of features manually extracted with signal pre-processing techniques and features adaptively extracted from raw data without pre-processing.

Manual features are extracted from vibrational signals using wavelet package transform with a maximum layer of 5 and db4. For each frequency-band, 19 features are obtained. In this way, the dimension of manual features for each vibration signal is 19 × 32. The average diagnostic accuracies of these diagnostic models are shown in Table 2.

From Table 2, we can find that diagnostic models based on BPNN and SVM are more sensitive to the forms of features inputted into the diagnostic model [41]. The average testing accuracies of these two shallow structured models with the input of manual features and features without pre-processing are 84.27% and 81.46%, 79.62% and 76.49%, respectively. However, the accuracies of the proposed model are very close, which are 94.23% and 93.84%. It indicates that without complex previous signal pre-processing, the proposed model can still recognize the vast majority of testing samples. It indicates the advantage of the proposed deep learning model on processing raw data adaptively and efficiently.

With the purpose of proving superiority of the proposed model combining MDNNs with multi-channel signal fusion, a series of experiments with the same sample set is implemented to compare some frequently used techniques in intelligent fault diagnosis. For each experiment, ten trials are implemented. The average training accuracy and testing accuracy of these different diagnostic models are given in Table 3. The average training time of all the diagnostic models are given in Table 4.

As shown in Table 3, the average training accuracy and testing accuracy of the proposed model with single-channel data is 91.42% and 90.17%. The average training accuracy and testing accuracy of the proposed model based on multiple DNNs with deep architecture for multi-channel feature learning are 95.84% and 92.76%, which are higher than other diagnostic models with different kinds of data. From the contrastive results in Table 3, it reveals that the capability significantly exceeds the diagnostic model based on SVM with shallow architecture for multi-channel feature learning, which are 81.28% and 77.63%. By contrast analysis, we can summarize that:
(1)In general, deep architecture of neural networks can effectively extract essential and useful features from raw data. However, it is hardly to obtain favorable results by using BPNN with deep architecture. Furthermore, as shown in Table 3, the variance of testing accuracy and training accuracy is obviously higher than other models. It indicates that the performance of BPNN is unstable due to the local minimum problem. The reason for this disadvantage is that the stability of BPNN will be affected by the initial value of network parameters [10]. It may also lead to obvious deviation in the procedure of error back propagation.(2)From Table 3, it shows that the deep learning method shows its obvious superiority in feature learning. However, even with multiple hidden layers and multi-channel data, the diagnostic performance of the model based on BPNN is still far from satisfactory, which are 84.56% and 81.14%. Feature learning with deep architecture contains a process of pre-training and a process of fine-tuning. The local minimum problem of traditional BPNN can be obviously solved through the optimization of initial weights layer-by-layer during the procedure of pre-training in unsupervised way and the adjustment of weights during the procedure of fine-tuning in supervised way, which are the typical characteristics of deep neural networks [49].(3)From the separation of typical features fused from the hidden layers shown in Figure 14, Figure 15 and Figure 16, we can find that a diagnostic model based on MDNNs can automatically extract fault-sensitive features that directly affect the final diagnostic results. In addition, it can adaptively mine the complex non-linear relevance between raw data and several fault modes, which are crucial for condition monitoring [29]. The capability of the model is independent on engineering experiences and prior knowledge in application area.

As shown in Table 4, the average training time of the proposed model with multi-channel data is 39.49 s. From the contrastive results in Table 4, we can find that the training time of the diagnostic model based on SVM with shallow architecture which is 5.93 s is obviously less than diagnostic models based on MDNNs and BPNN with deep architecture. The training time of the proposed model is the largest, which is 58.56 s.

The computational complexity of the proposed model is a typical shortcoming when the number of hidden layers increased. The reason is that the number of weights of shallow architecture that should be adjusted is much less than deep architecture containing multiple hidden layers [47]. Though the training time of the proposed model is the longest, the training process is accomplished within one minute.

Moreover, compared with the training time of models without fusion strategy, which is 58.56 s, the training time of models with the fusion of multi-channel data using the LPP algorithm is obviously decreased to improve the training efficiency. The reason behind the improvement is that the fusion process using LPP can decrease the dimension of high-dimensional features so as to decrease the computational complexity [43,44].

## 5. Conclusions

In this article, a deep learning method for multi-channel sensory data combining a deep neural network for feature mining and feature fusion is proposed. First, a deep structure MDNNs made up of multiple auto-encoders is constructed to adaptively extract significant features from sensory signals and to mine complex relation between symptoms and fault patterns. Secondly, LPP is used to fuse representative deep features learned from multi-channel sensory data using MDNNs. Finally, an intelligent diagnostic model is constructed by inputting the fused deep features into softmax.

The diagnostic model is employed into the application of intelligent failure recognition for automobile final drive. The average training accuracy and testing accuracy of the proposed model based on multi-channel data are 95.84% and 92.76%, which are higher than other contrastive models. Compared with fault recognition with single-channel signals, the proposed method can effectively enhance the capability of failure classification from 90.17% to 92.76%. Moreover, the variance of the training accuracy and testing accuracy is obviously the smallest among these contrastive models. Through the contrastive analysis of experimental results, it proves that the proposed diagnostic model is more superior and stable than other models in fault recognition and condition monitoring. The deep structure of feature learning from multi-channel data can effectively solve the limitation of a single sensor and the adaptive fusion of feature can avoid the heterogeneity and redundancy of deep features extracted from multi-channel data. This research is instructive to the technology of industry. In the future, the authors would continue to investigate the intelligent diagnosis using multi-channel data in different types so as to further enhance the diagnostic accuracy and efficiency.

## Figures and Tables

**Figure 1 sensors-20-04300-f001:**
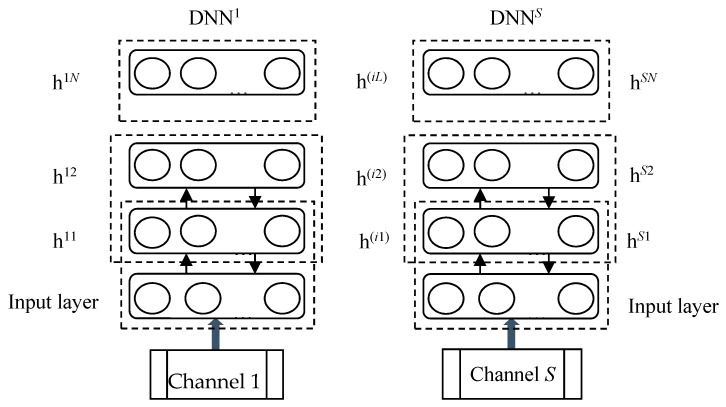
Architecture of multiple deep neural networks (MDNNs).

**Figure 2 sensors-20-04300-f002:**
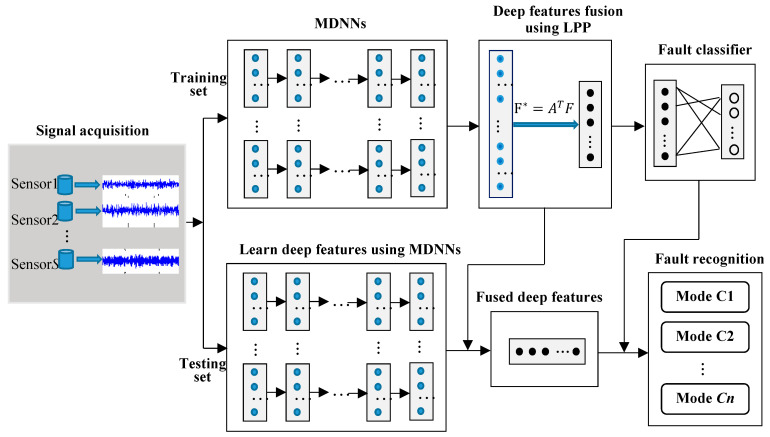
Procedure of the proposed model.

**Figure 3 sensors-20-04300-f003:**
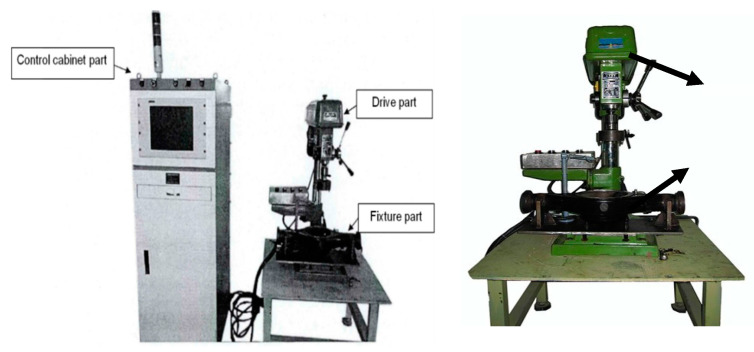
Test rig of automobile final drive.

**Figure 4 sensors-20-04300-f004:**
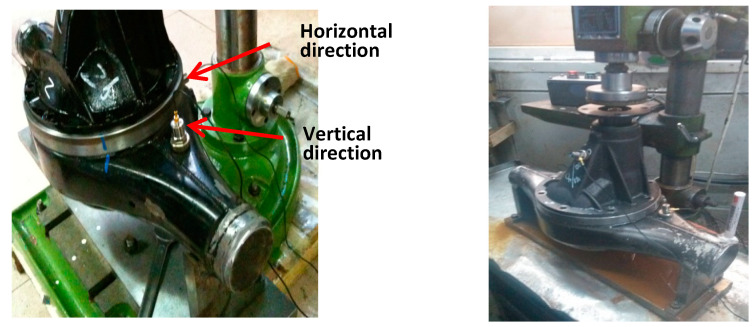
Installation of multiple sensors.

**Figure 5 sensors-20-04300-f005:**
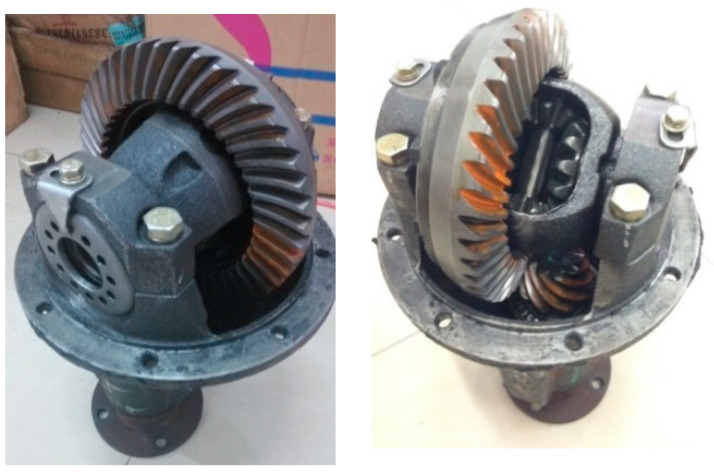
Pivotal part in final drive: gear pair.

**Figure 6 sensors-20-04300-f006:**
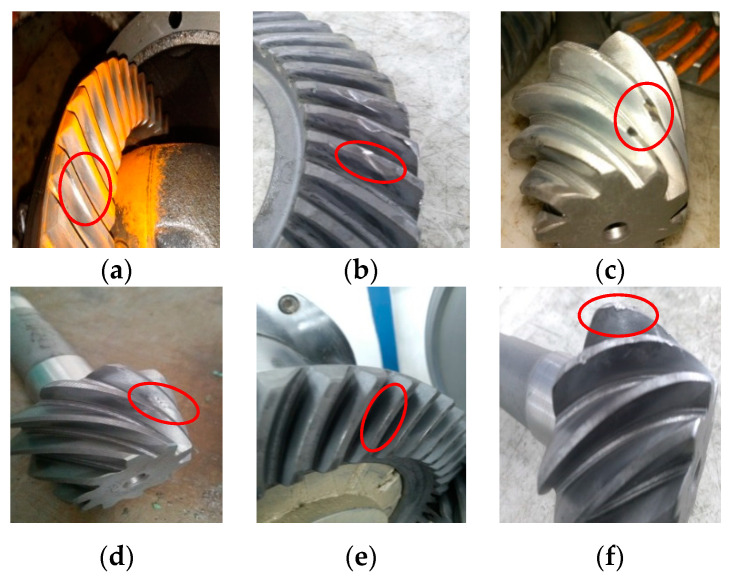
Faulty gears pairs: (**a**) gear crack, (**b**) gear error, (**c**) gear tooth broken, (**d**) gear burr, (**e**) misalignment, (**f**) gear hard point.

**Figure 7 sensors-20-04300-f007:**
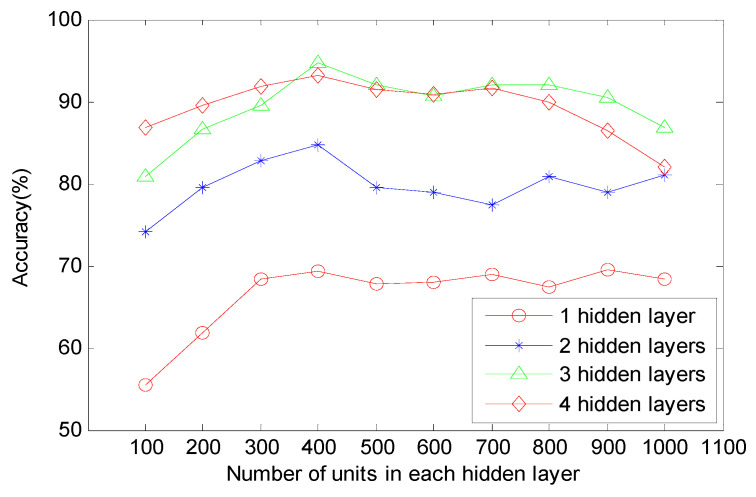
Performances of different deep architectures.

**Figure 8 sensors-20-04300-f008:**
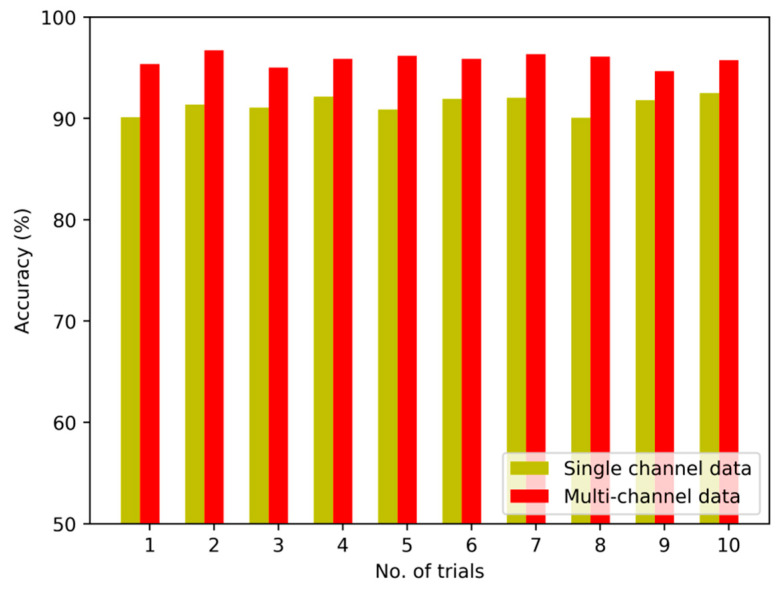
Diagnostic results of 10 trials for the proposed method.

**Figure 9 sensors-20-04300-f009:**
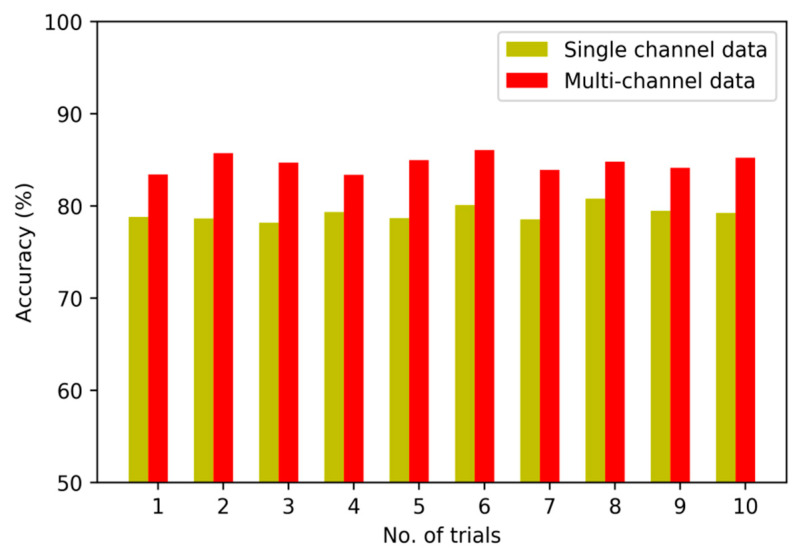
Diagnostic results of 10 trials for BPNN-based model.

**Figure 10 sensors-20-04300-f010:**
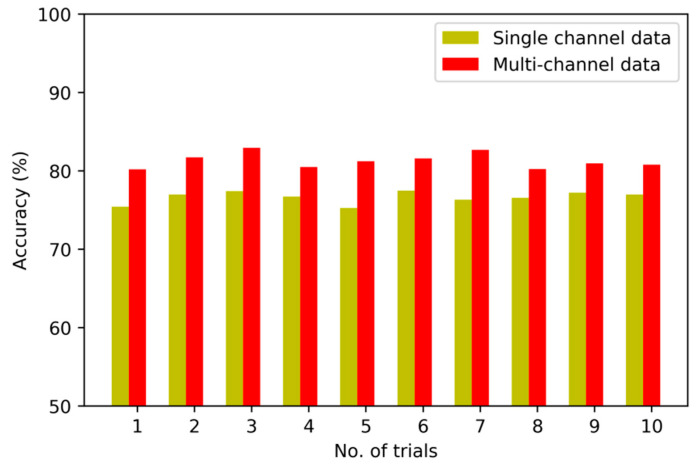
Diagnostic results of 10 trials for Support Vector Machine (SVM)-based model.

**Figure 11 sensors-20-04300-f011:**
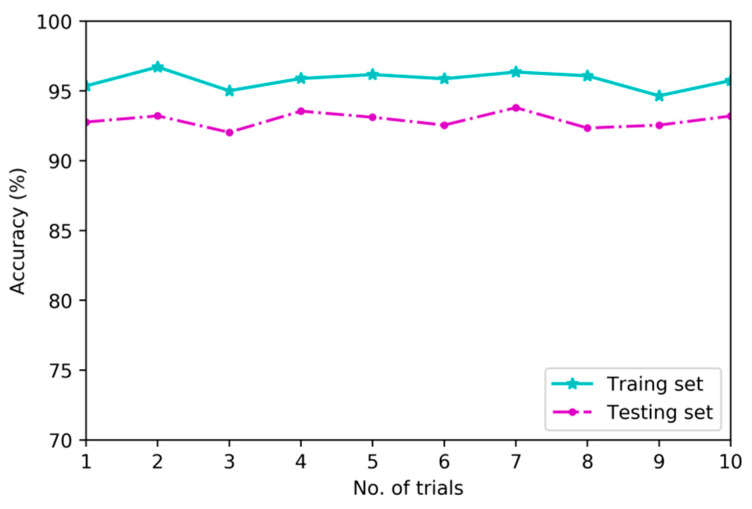
Classification results of MDNNs.

**Figure 12 sensors-20-04300-f012:**
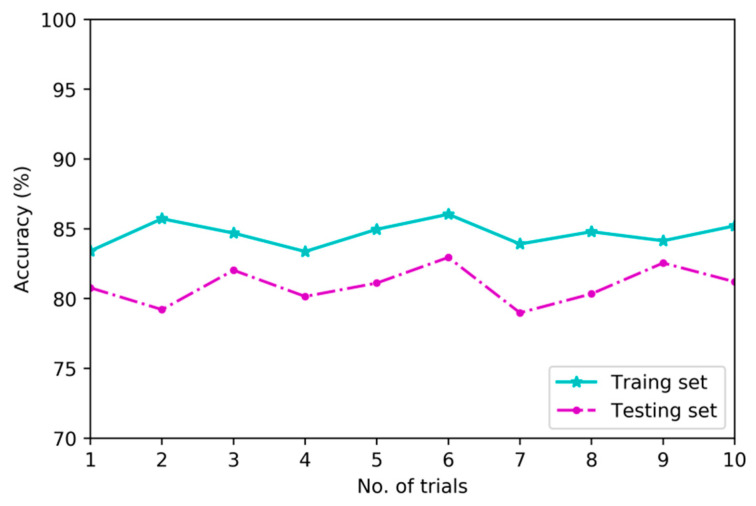
Classification results of BPNN.

**Figure 13 sensors-20-04300-f013:**
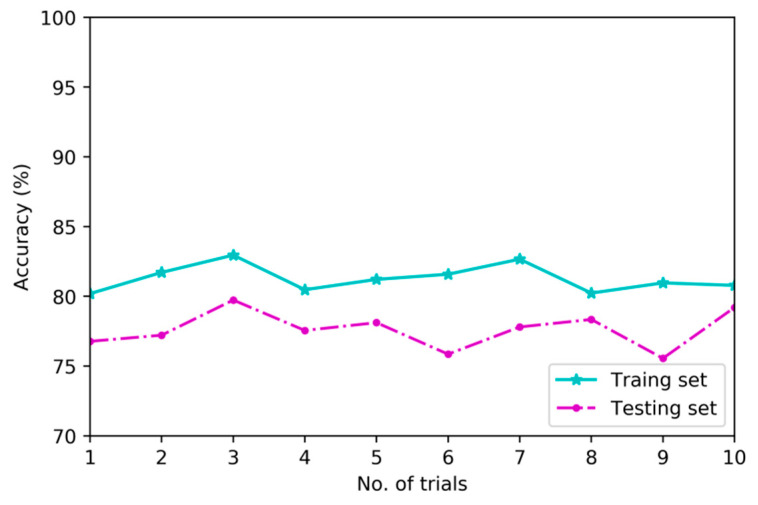
Classification results of SVM.

**Figure 14 sensors-20-04300-f014:**
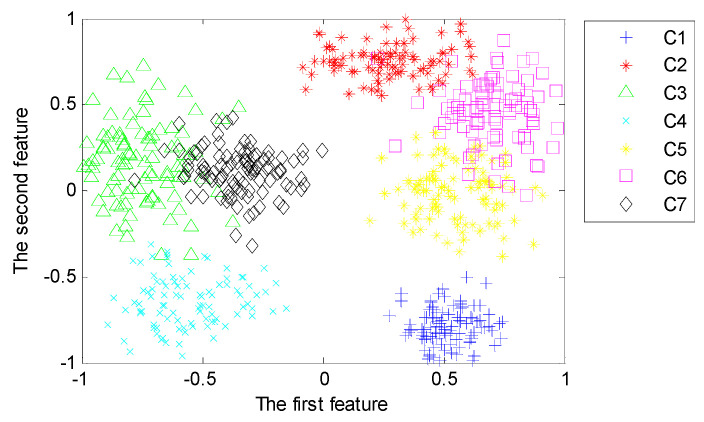
Typical features fused from the first hidden layer using Locality Preserving Projection (LPP).

**Figure 15 sensors-20-04300-f015:**
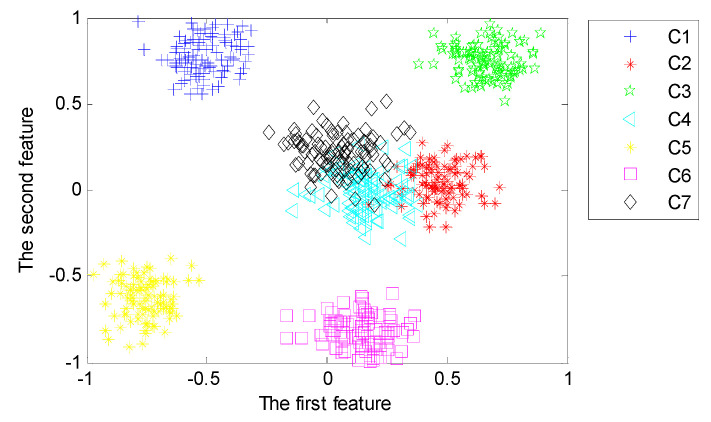
Typical features fused from the second hidden layer using LPP.

**Figure 16 sensors-20-04300-f016:**
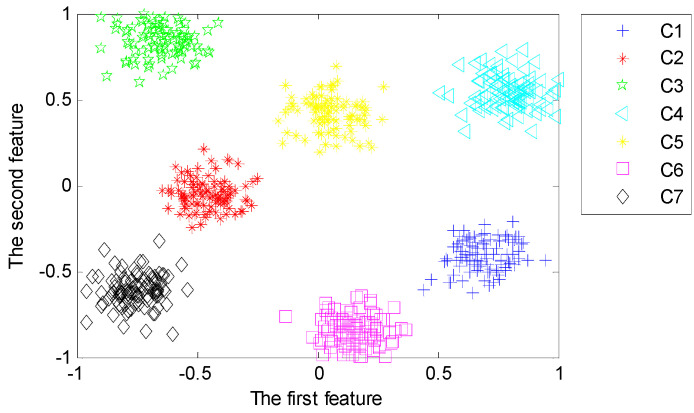
Typical features fused from the third hidden layer using LPP.

**Figure 17 sensors-20-04300-f017:**
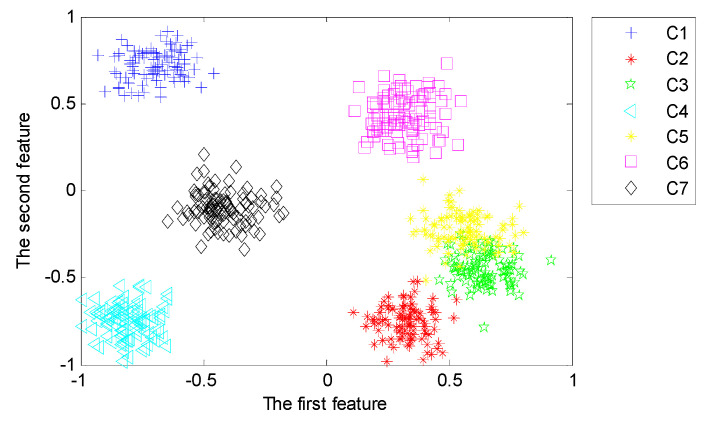
Principal components fused by Principle Component Analysis (PCA).

**Table 1 sensors-20-04300-t001:** Fault modes’ description.

Label	Fault Patterns	Size of Training Set	Size of Testing Set
C1	Normal status	1400	350
C2	Gear crack	1400	350
C3	Gear error	1400	350
C4	Gear tooth broken	1400	350
C5	Gear burr	1400	350
C6	Misalignment	1400	350
C7	Gear hard point	1400	350

**Table 2 sensors-20-04300-t002:** Average accuracy of different diagnostic models.

Models	Average Testing Accuracy (%)	
	Features Manually Extracted with Signal Pre-Processing	Features Adaptively Extracted without Pre-Processing
The proposed model	94.23 (±1.54)	93.84 (±0.73)
Model based on BPNN	84.27 (±3.81)	81.46 (±4.32)
Model based on SVM	79.62 (±2.88)	76.49 (±2.92)

**Table 3 sensors-20-04300-t003:** Average accuracy of different diagnostic models.

Models	Average Training Accuracy (%)		Average Testing Accuracy (%)	
	Single-Channel Data	Multi-Channel Data	Single-Channel Data	Multi-Channel Data
The proposed method	91.42 (±1.11)	95.84 (±0.94)	90.17 (±1.29)	92.76 (±1.13)
Model based on BPNN	79.09 (±4.59)	84.56 (±3.72)	76.29 (±4.80)	81.14 (±4.05)
Model based on SVM	76.62 (±2.97)	81.28 (±2.63)	74.32 (±3.47)	77.63 (±3.06)

**Table 4 sensors-20-04300-t004:** Average training time of different diagnostic models.

Models	Average Training Time (s)	
	Without Fusion Multi-Channel Data	Fusion Multi-Channel Data
The proposed model	58.56	39.49
Model based on BPNN	30.39	18.61
Model based on SVM	10.22	5.93

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
