# Peer review of "A Deep Learning Model for Fault Diagnosis with a Deep Neural Network and Feature Fusion on Multi-Channel Sensory Signals"

_sensors, 2020, doi:10.3390/s20154300_

Round 1

Reviewer 1 Report

They claim the deep learning method for multi-channel sensory data combining deep neural network for feature mining and feature fusion. The work suffers from some issues:

  1. No recall of the model reported. please explain this in the model and also provide some related results for it.
  2. The complexity of the proposed method did not well explore. Also, how can we manage the multi-layer demands instantiated as a time served sample heterogeneous combined data? it is not addressed for the paper.
  3. The background is limited and expected to include most recite technologies like ‘A Dynamic Membership Data Aggregation (DMDA) Protocol for Smart Grid’.
  4. Try to use higher quality plots, using png or pdf format for the plots: results and architecture.

Author Response

Dear Reviewer,

       I really appreciate the editor and reviewer for the patient guidance and valuable suggestions from which I really benefited a lot. According to the comment, I made some modification and added more detailed description to improve the logicality.

Best wishes.

Qing Ye

Reviewer 2 Report

Line 24: how? Insert the exact results here.

Line 25: insert validation method and limitations.

Line 71: why is this proposed? Justification is required.

The introduction section needs to be improved by inserting: the value of this study, contributions, and novelty. The last paragraph should outline the content of the paper.

Line 209: the data profile needs to be clarified.

Line 232: why 10 simulation is enough?

 Line 336: insert details of PCA.

Line 349: the discussion section should be improved by comparing the outcome and observations with the literature.  What is the current understanding and state of the art, and what you offer? What are the limitations and future studies? This section has no reference at present, so the author needs to use the references used in the introduction and some new ones and discuss the results here.

Line 360: strong justification and validation required (to be discussed here) since there is a huge difference between the results of different methods (70% to 90%).

This is a big claim and so needs a strong validation discussion citing fresh articles.

Line 364 to 378, these sections need to be referred to as exact results, tables, and figures when claiming or describing something.

Line 390: for each one, you need to insert the results, validation, limitation, and future study/step.

Line 396: how proves? Exact information required

Line 399: the authors' plan for the future is not necessary. Future studies based on the limitations of this experimentation need to be clarified.

Line 394: the novelty and contribution of this paper should be clearly mentioned here.

Author Response

(The authors gave the same response as above.)

Reviewer 3 Report

The article deals with the detection/classification of rotating machines through the aid of data fusion and machine learning. The topic and studied component (gearbox) are always of interest to researchers and industrial practitioners. The theoretical foundation for the applied approaches were adequately described. The results were also clearly presented. However, the following must be adequately addressed for this to be acceptable:

  1. There is need for extensive proofreading to eliminate current grammatical errors in several parts of the article.
  2. considering the maturity of rotating machines faults diagnosis, data fusion, machine learning, PCA and gearboxes, the review of existing studies conducted here is quite shallow and must be expanded. For instance, there are immense amount of studies on the combination of PCA, various machine learning techniques and data fusion, which may be arguably be simplified as well but not mentioned here. It is therefore advised that the authors include such articles as well as explain how the current differs. Such studies are not limited to but include:
    1. Yunusa-Kaltungo A, Sinha JK, Nembhard AD. A novel fault diagnosis technique for enhancing maintenance and reliability of rotating machines. Structural Health Monitoring. 2015 Nov;14(6):604-21.
    2. Jing L, Wang T, Zhao M, Wang P. An adaptive multi-sensor data fusion method based on deep convolutional neural networks for fault diagnosis of planetary gearbox. Sensors. 2017 Feb;17(2):414.
    3. Lu Y, Tang J, Luo H. Wind turbine gearbox fault detection using multiple sensors with features level data fusion. Journal of Engineering for Gas Turbines and Power. 2012 Apr 1;134(4).
    4. Nembhard AD, Sinha JK, Yunusa-Kaltungo A. Development of a generic rotating machinery fault diagnosis approach insensitive to machine speed and support type. Journal of Sound and Vibration. 2015 Feb 17;337:321-41.
    5. Liu Z, Guo W, Tang Z, Chen Y. Multi-sensor data fusion using a relevance vector machine based on an ant colony for gearbox fault detection. Sensors. 2015 Sep;15(9):21857-75.
    6. Li C, Sanchez RV, Zurita G, Cerrada M, Cabrera D, Vásquez RE. Gearbox fault diagnosis based on deep random forest fusion of acoustic and vibratory signals. Mechanical Systems and Signal Processing. 2016 Aug 1;76:283-93.
    7. Khazaee M, Ahmadi H, Omid M, Banakar A, Moosavian A. Feature-level fusion based on wavelet transform and artificial neural network for fault diagnosis of planetary gearbox using acoustic and vibration signals. Insight-Non-Destructive Testing and Condition Monitoring. 2013 Jun 1;55(6):323-30.
    8. Khazaee M, Ahmadi H, Omid M, Moosavian A, Khazaee M. Classifier fusion of vibration and acoustic signals for fault diagnosis and classification of planetary gears based on Dempster–Shafer evidence theory. Proceedings of the Institution of Mechanical Engineers, Part E: Journal of Process Mechanical Engineering. 2014 Feb;228(1):21-32.
    9. Vanraj, Dhami SS, Pabla BS. Hybrid data fusion approach for fault diagnosis of fixed-axis gearbox. Structural Health Monitoring. 2018 Jul;17(4):936-45.
    10. Yunusa-Kaltungo A, Cao R. Towards Developing an Automated Faults Characterisation Framework for Rotating Machines. Part 1: Rotor-Related Faults. Energies. 2020 Jan;13(6):1394.
  3. Further details about the experimentally simulated conditions and the selected signal processing parameters must be provided. For instance, sata was sampled at 12kHz. How was this sampling frequency determined? Was this based on pre-estimated gear mesh frequencies? 
  4. It was also stated that condition monitoring sensors were installed at the appropriate locations. What are these appropriate monitoring locations? What kind of sensors were applied and what are their characteristics/specifications especially linearity.

Please address the following points accordingly.

Author Response

(The authors gave the same response as above.)

Reviewer 4 Report

In this paper, an MDNN network is proposed for feature extraction of multi-channel sensor signals. After that, The Locality Preserving projection method is adopted to fuse the multi-feature signals, and finally, the fusion features are classified by using the full connection network. Experiments show that the proposed multi-channel signal fusion fault diagnosis framework is advanced. The work is relatively complete and has certain academic value. However, In my opinion, there are still several problems in the article:

1.In the introduction section, please add the literature review of the current application of multi-channel and multi-sensor fusion algorithm in the field of fault diagnosis.

2.The main object of this paper is multi-channel sensory signals, but only 2 channels signals are used in the experiment, so the number of channels of input signals should be increased for experimental verification.

3.In Section 3.1, please discuss the necessity of fine tuning of the mentioned network and detail the fine tuning process.

4.In Section 3.3, please add the detailed parameters of the proposed network structure, including the number of network layers and neuron nodes of each module.

5.In Section 4.1, what is the rotational speed corresponding to the training data and test data collected in the experiment? What is the variation of the rotational speed?

Author Response

(The authors gave the same response as above.)

Reviewer 5 Report

To the Authors:

Hereby, After the reviewing process. I sending my comments of the paper: “A Deep Learning Model for Fault Diagnosis with Deep Neural Network and Feature Fusion on Multi-channel Sensory Signals”.

A deep learning method with deep architecture and feature fusion on multi-channel signals were developed by the authors. It includes the assembly of deep architecture for feature learning, fusion of extracted from multi-channel sensors data and designing of intelligent diagnostic model using softmax classifier.

The authors could have a positive impact in the field of machine intelligent diagnosis, with the novelty of this research could the combination of deep learning models with feature extraction of multi channels, however, the authors need to explain deeply all the new findings in details. For example, what is the criteria of the improved feature extraction with multi channels? How about the extracted features are in time domain or frequency domain.?. In my opinion, it`s necessary to explain further, how it was improved the diagnosis precision, not only mention that different signals have different characteristics.

There is another aspect with the use of Locality Preserving Projections (LPP) for dimensionality reduction. It was also used Principal component analysis, both have same characteristics, what was the idea to use both?

Author Response

(The authors gave the same response as above.)

Round 2

Reviewer 1 Report

The paper is not easy to read, the results and the related explanations are solid and the can be published.

Author Response

Dear reviewer,

I appreciate the reviewer for the patient guidance and valuable suggestions from which I benefited a lot.

I really have a lot of space for progress in writing English papers. I will keep going and do my best.

Finally, thanks again for the editor and reviewer’s patience to improvement the overall level of this article. I really have benefited a lot during the process of revision.

Please accept my best wishes for your happiness and success.

Qing Ye

Reviewer 2 Report

The conclusion still needs to be imrpved. Line 462, what superior and more than others refers to? can be specific how much more? This is specifically important since it can be the contribution of the paper. Specific numerical limitations, results and contributions of the paper should be clarified in the conclutions.

Author Response

Dear reviewer,

I really appreciate the reviewer for the patient guidance and valuable suggestions from which I really benefited a lot.

The replies to comments and details of changes are in the attachment.

Best wishes.

Qing Ye
